# Auditory or Audiovisual Stimulation Ameliorates Cognitive Impairment and Neuropathology in ApoE4 Knock-In Mice

**DOI:** 10.3390/ijms24020938

**Published:** 2023-01-04

**Authors:** Harry Jung, Yeonkyeong Lee, Sang-Hwa Lee, Jong-Hee Sohn

**Affiliations:** 1Institute of New Frontier Research Team, College of Medicine, Hallym University, Chuncheon 24252, Republic of Korea; 2Department of Neurology, Chuncheon Sacred Heart Hospital, Hallym University College of Medicine, Chuncheon 24252, Republic of Korea

**Keywords:** hearing loss, Alzheimer’s disease, auditory, visual, ApoE4

## Abstract

We hypothesized that auditory stimulation could reduce the progression of Alzheimer’s disease (AD), and that audiovisual stimulation could have additional effects through multisensory integration. We exposed 12 month old Apoe^tm1.1(APOE*4)Adiuj^ mice (a mouse model of sporadic AD) to auditory (A) or audiovisual stimulation (AV) at 40 Hz for 14 days in a soundproof chamber system (no stimulation, N). Behavioral tests were performed before and after each session, and their brain tissues were assessed for amyloid-beta expression and apoptotic cell death, after 14 days. Furthermore, brain levels of acetylcholine and apoptosis-related proteins were analyzed. In the Y-maze test, the percentage relative alternation was significantly higher in group A than in group N mice. Amyloid-beta and TUNEL positivity in the hippocampal CA3 region was significantly lower in group A and group AV mice than in group N mice (*p* < 0.05). Acetylcholine levels were significantly higher in group A and group AV mice than in group N mice (*p* < 0.05). Compared to group N mice, expression of the proapoptotic proteins Bax and caspase-3 was lower in group A, and expression of the antiapoptotic protein Bcl-2 was higher in group AV. In a mouse model of early-stage sporadic AD, auditory or audiovisual stimulation improved cognitive performance and neuropathology.

## 1. Introduction

Alzheimer’s disease (AD) is a neurodegenerative disease that causes a progressive loss of cognition. While many environmental and genetic risk factors are associated with the development of AD, its etiology is poorly understood. Among the observed changes in the brains of AD patients is the abnormal accumulation of amyloid-beta (Aβ), either extracellularly as amyloid plaques and tau proteins or intracellularly as neurofibrillary tangles. Both have been implicated in impaired neuronal functioning and connectivity, resulting in a progressive loss of brain function [1,2]. Currently, there are only two classes of approved drugs to treat AD: cholinesterase inhibitors and N-methyl D-aspartate antagonists. However, these are effective only in treating the symptoms of AD; they do not prevent the disease or stop or reverse its progression [2,3].

The sensory deterioration that occurs in AD, including olfactory impairment, visual impairment, and hearing loss, is often faster than that in normal aging [4,5,6,7]. Conversely, sensory loss increases the risk of dementia, especially AD, and cognitive dysfunction in older adults [8,9,10]. In particular, an association among age-related cognitive decline, AD, and age-related hearing loss has been reported in epidemiological studies that controlled for potentially confounding variables such as age, sex, apolipoprotein Eε4 status, and cardiovascular risk factors [11,12,13,14,15,16,17,18]. Additionally, in studies using animal models, sensorineural hearing loss induced by noise overexposure suppressed hippocampal neurogenesis, with the animals also exhibiting cognitive decline [19,20]. A study of a model of conductive hearing loss in AD mice also found that partial hearing loss may exacerbate memory impairment in AD [21].

Given the failure of many drugs to target the pathology of AD [22] and the inadequacy and significant side-effects of the drugs currently in use, greater attention has been paid to nonpharmacologic interventions [23,24,25]. These include common noninvasive stimulation, such as sensory, transcranial magnetic, and transcranial electrical (direct and alternating current) stimulation [26,27,28,29]. Sensory-based interventions are noninvasive and easy to achieve, do not cause side-effects, and are inexpensive, making them a promising approach to the treatment of AD [26].

In clinical trials, rhythmic auditory stimulation in patients with mild to moderate AD or mild cognitive impairment was shown to increase cognitive performance and improve anxiety [30,31]. In studies of animal models of AD, auditory stimulation was shown to reduce Aβ and tau pathology and to improve cognitive ability [32,33]. These results support the need for further research into the therapeutic effect of auditory stimulation in human AD patients.

We hypothesized that auditory stimulation could reduce the progression of cognitive decline and ameliorate the pathology of AD; moreover, we hypothesized that, through multi-sensory integration, a combination of auditory and visual stimulation could further enhance cognition. Previous studies of AD used a transgenic mouse model, APP, presenilin (PSEN1 or PSEN2), or their combinations (3 × Tg, 5 × Tg) [32,33]. Despite valuable insights into the mechanisms of AD obtained with these models, they more likely mimic the genetic (familial) form of the disease. However, as early-onset familial AD accounts for <5% of AD cases, mouse models that mimic the much more common, late-onset progression seen in sporadic human AD are needed [34,35,36]. Although sporadic AD has multiple causes, the strongest genetic risk factor in humans is the apolipoprotein Eε4 allele. Thus, in this study we used an ApoE4 transgenic knock-in mouse model of late-onset sporadic AD [37,38,39] to investigate the effects of auditory or audiovisual stimulation on cognition and neuropathology in an ApoE4 knock-in mouse, as a clinically relevant model of sporadic AD.

## 2. Results

### 2.1. Improvement in Cognitive Function of ApoE4 KI Mice after Sensory Stimulation

Spatial learning and the memory ability of ApoE4 KI mice were examined using the MWM test. Distance moved, velocity (cm/s), time moving (%), time not moving (%), time in quadrant (%), absolute escape latency, and relative escape latency (ratio) were measured. In the MWM test, the relative escape latency of the A group was less than that of the N group, but the difference was not significant. In the two-way ANOVA based on the sensory stimulation data, there were no statistically significant differences in other parameters of the MWM (Appendix A).

A YM test was performed to assess short-term spatial working memory based on spontaneous alternation. The absolute and relative percent alternation was measured. A two-way ANOVA based on sensory stimulation revealed significant effects of time on absolute (times, F = 11.59, *p* < 0.001; groups, F = 2.111, *p* = ns; interactions, F = 4.116, *p* < 0.001) and relative (times, F = 4.055, *p* < 0.05; groups, F = 1.228, *p* = ns; interactions, F = 3.071, *p* < 0.05) alternation (Table 1). Post hoc pairwise comparisons showed a significantly lower absolute alternation on post-stimulation day 7 in the AV group (45.90 ± 7.78 vs. 80.77 ± 5.00, *p* < 0.05) than in the N group. On post-stimulation day 14, absolute (71.46 ± 8.10 vs. 43.17 ± 1.67, *p* < 0.05) and relative (1.20 ± 0.14 vs. 0.62 ± 0.02, *p* < 0.05) alteration was significantly higher in the A group than in the N group (Figure 1).

### 2.2. Reduction in Aβ Deposition in the Hippocampus of ApoE4 KI Mice after Sensory Stimulation

Figure 2 shows the Aβ expression levels in ApoE4-KI mice after auditory and audiovisual stimulation. In the one-way ANOVA, the Aβ expression levels of the hippocampal cornu ammonis (CA) 3 region were significantly lower in the A group (1174.67 ± 457.19 vs. 3282.00 ± 338.95) and AV group (1348.00 ± 263.17 vs. 3282.00 ± 338.95) than in the N group, as also shown by the post hoc pairwise tests (*p* < 0.01). However, this difference was not observed in other brain regions, including hippocampal CA 1 and 2, dentate gyrus, and cortex (Table 2).

### 2.3. Reduction in Apoptosis in the Hippocampus of ApoE4 KI Mice after Sensory Stimulation

TUNEL staining was used to observe the effect of sensory stimulation on apoptosis in hippocampal and cortical cells. Figure 3 shows the TUNEL positivity of the cells after the auditory and audiovisual stimulation of ApoE4-KI mice. In the one-way ANOVA, TUNEL positivity in the hippocampal CA3 region was significantly lower in the A group (0.11 ± 0.03 vs. 0.39 ± 0.16) and AV group (0.13 ± 0.03 vs. 0.39 ± 0.16) than in the N group, as also shown in post hoc pairwise tests (*p* < 0.01). By contrast, this difference was not observed in other brain regions, including hippocampal CA 1 and 2, dentate gyrus, and cortex (Table 3).

### 2.4. Changes in Brain Acetylcholine Levels in ApoE4 KI Mice after Sensory Stimulation

Total choline, free choline, and Ach levels differed significantly in the three groups of mice subjected or not to auditory and audiovisual stimulation. The results of the ELISA showed significantly higher total choline, free choline, and Ach levels in the A and AV groups than in the N group (Table 4, *p* < 0.001), whereas the differences between the A and AV groups were not significant (Figure 4).

### 2.5. Changes in Apoptosis-Related Proteins in ApoE4 KI Mice after Sensory Stimulation

The Western blot bands and relative expression levels of the apoptosis-related proteins Bax, Bcl-2, and caspase-2 and of the β-actin control in ApoE4-KI mice subjected to auditory and audiovisual stimulation are presented in Figure 5. The original Western blot images used to analyze the proteins are shown in the Appendix A. In the Western blot, the expression of proapoptotic proteins was significantly lower in the A group than in the N group (Bax, 0.64 ± 0.06 vs. 0.99 ± 0.10, *p* < 0.01; caspase-3, 1.20 ± 0.10 vs. 1.48 ± 0.08, *p* < 0.05). The expression of Bax was significantly lower in the AV group than in the N group (0.64 ± 0.06 vs. 0.97 ± 0.11 vs., *p* < 0.01), while the expression of the antiapoptotic protein Bcl-2 was significantly higher (1.96 ± 0.19 vs. 1.49 ± 0.13, *p* < 0.05) (Table 5).

## 3. Discussion

In this study, the effects of auditory and audiovisual stimulation on cognition and neuropathology in ApoE4-KI mice, an experimental model of sporadic AD, were investigated. In the YM test, used to assess short-term and spatial working memory, the A group performed significantly better than the N group. In the histological analysis, both Aβ expression and TUNEL positivity in the hippocampal CA3 region were significantly lower in the A and AV groups than in the N group. Measurements of total choline, free choline, and Ach levels showed significantly higher levels in the A and AV groups than in the N group. Expression of the proapoptotic proteins Bax and caspase-3 was significantly lower in the A group, and that of the antiapoptotic protein Bcl-2 significantly higher in the AV group than in the N group.

To date, most of the evidence of the beneficial effects of auditory stimulation on cognition or behavioral performance has come from clinical rather than animal studies [40]. In nondemented older adults and in patients with amnestic mild cognitive impairment, auditory stimulation applied during sleep had positive effects on morning word recall [41,42]. Auditory stimulation combined with visual stimuli, both delivered at 40 Hz, was shown to improve cognitive performance, awareness of surroundings, interaction discussion/storytelling abilities, and alertness in AD patients [31]. In an animal study, spatial memory and recognition in 5x-FAD mice were improved after 40-Hz auditory stimulation, assessed using behavioral tests, including MWM, novel object recognition, and novel object location tasks [33]. In studies of 5x-FAD and tauopathy mouse models, auditory stimulation at 40 Hz or gamma oscillations reduced amyloid levels [32,33]. Auditory stimulation combined with light-induced gamma oscillations, but neither alone, produced microglial clustering responses and decreased the amyloid content in the medial prefrontal cortex of the mice. These results suggest that multiple sensory modalities, with wide-ranging effects across multiple brain areas, can improve cognitive function [33]. Similarly, our study showed an improvement in cognitive function and neuropathology after gamma-frequency auditory stimulation in ApoE4 KI mice; however, audiovisual stimulation of the mice did not yield better results than auditory stimulation alone.

Most mouse models used to develop AD therapies are based on familial AD, triggered by mutations in APP, PSEN1, or PSEN2, but most AD patients have late-onset sporadic AD. Our study, therefore, examined ApoE4 KI mice, a mouse model of sporadic AD. Strong risk factors for this form of AD are the apolipoprotein Eε4 allele along with rare point mutations in the triggering receptor expressed on myeloid cells 2 (TREM2) [43]. The use of validated, new humanized APOEε4/ε4 mouse strains of clinical relevance was an advantage of this study [44].

In our study, Aβ expression levels in the hippocampal CA3 region were significantly lower, and markers of the cholinergic system were at significantly higher levels, in the A and AV groups than in the N group. TUNEL positivity in the hippocampal CA3 region was significantly lower in the A and AV groups than in the N group, similar to the results of the biochemical analyses of apoptosis-related proteins. A previous study found a moderate correlation between TUNEL-positive cells and amyloid plaque density in AD patients and suggested that Aβ is one of multiple factors provoking cell injury in AD [45]. The deposition of Aβ in the brain is a well-established central link in the pathological changes of AD. Moreover, the pathological processes triggered by Aβ deposition promote further deposition, leading to a response cascade. It has also been reported that Aβ is neurotoxic, thus inducing nerve cell apoptosis and, in turn, impairing cognitive function [46,47]. Abnormalities in the cholinergic system in AD also induce abnormal amyloid precursor protein metabolism and tau phosphorylation, resulting in neurotoxicity, neuroinflammation, and apoptosis [48,49]. Our results suggest that auditory or audiovisual stimulation affects the cholinergic system, thereby reducing amyloid accumulation and suppressing apoptosis and neuronal injury.

In a previous study of a mouse model of AD using 5x-FAD and APP/PS1 mice, visual stimulation consisting of a 40 Hz light-flickering regimen induced gamma oscillations in addition to reducing the levels of Aβ and microglia transformation [50]. Combined auditory and visual stimulation using light flickering at 40 Hz reduced Aβ pathology in the hippocampus, auditory cortex, and medial frontal cortex of 5x-FAD mice, whereas this was not the case when either stimulus was used alone [33]. In a mouse model of neurodegeneration, a combination of flickering light and sound stimuli with 40 Hz tones not only reduced Aβ plaques but also had farther-reaching effects extending to higher-order brain areas, including the hippocampus and prefrontal cortex [51]. As demonstrated by these and other studies, the positive effects on cognitive performance and neuropathological findings were achieved with auditory, visual, or combined stimulation delivered at 40 Hz, i.e., gamma frequency. Gamma waves (~25–100 Hz) are the fastest brain waves in the human brain and play an important role in cognitive function, particularly memory. They are also fundamental to intra-brain communication and healthy brain activity. Abnormal gamma oscillations are a feature of AD both in mouse models and human patients. Gamma entrainment through auditory or visual stimulation in mouse models of AD was shown to attenuate the pathology effectively and improve cognitive performance [52]. The stimulation settings used for animals in our study, i.e., 40 Hz, 2 h per day for 2 weeks, were similar to the auditory or audiovisual stimulation settings used in previous human studies [40]. In a clinical study, auditory stimulation delivered at 40 Hz (30 min twice a week over a 6 week period) improved cognitive performance in AD patients [31]. In a preliminary clinical trial, gamma flicker using audiovisual stimulation (40 Hz, 1 h daily over 4 or 8 weeks) strengthened functional connectivity and altered brain immune factors in patients with mild cognitive impairment related to AD, as evidenced by resting state EEG, neural circuits via default mode connectivity in fMRI, immune signaling via changes in cytokines, and immune factors in the CSF [53].

However, contrary to our hypothesis, audiovisual stimulation did not yield better results in the behavioral tests or with respect to neuropathological findings than obtained with auditory stimulation alone. The differences between our study and those discussed above can be attributed to differences in the AD mouse model, the age of the mice, and the stimulation protocols, including the duration of the sensory stimulus. In particular, in our study, the audio and visual stimuli were temporally congruent. The ability to synthesize information across multiple senses, referred to as multisensory integration, is modulated by both bottom-up stimulus features, such as the temporal structure of unimodal sensory stimuli, and top-down processes, such as attention. In previous studies, the response to two unimodal stimuli delivered closely in time was greater than simply the sum of those responses, and the magnitude of the multisensory enhancement decreased as the temporal asynchrony of the paired stimuli increased [54,55]. In our study, the audiovisual stimuli were temporally congruent, presented as a pair. However, a recent study showed that the time it takes for auditory and visual signals arising simultaneously from a common source to reach multisensory neurons in the brain is influenced by neural and nonneural factors; since sound travels much slower than light, it arrives later [56]. Other studies have shown that, as the simulated source distance increases, the alignment of audio and visual stimuli requires a delay in the delivery of the auditory stimulus by an amount corresponding to the additional sound travel time [57,58,59]. In the presence of cognitive dysfunction, such as attention deficit or high-level decision-making conflicts, bimodal sensory stimuli are perceived as simultaneous visual and auditory information rather than being processed as multisensory objects that enhance performance [60,61]. Therefore, the desired multisensory integration effect following audiovisual stimulation may not have been possible, due to the method of audiovisual stimulation and the use of a mouse model of cognitive dysfunction. Additional studies based on more precise research methods are needed to investigate the effect of multisensory integration by audiovisual stimulation.

As noted above, many of the previous AD studies used a mouse model of early-onset familial AD, whereas we used ApoE4-KI mice, an animal model of the more common late-onset sporadic AD. ApoE4-KI mice exhibit age-dependent impairments in hippocampus-dependent learning and memory commencing at 12 months and evident at 15–18 months [62,63,64]. The pathological changes in the 12 month old ApoE4-KI mice used in this study were consistent with early-stage disease; however, neuropathological abnormalities, mainly in the CA3 region of the hippocampus, were apparent. Some areas of the brain in AD are particularly vulnerable to specific degenerative processes that lead to neuronal dysfunction in the earliest stage of the disease, such as the prominent decreases in hippocampal neuronal density in the CA1 and CA3 regions of the hippocampus [65]. During AD progression, the oxidative stress pathway is activated in the CA3 region, progresses to the CA1 region, and then continues to other hippocampal and cortical areas [66]. The CA3 region of the hippocampus plays important roles in encoding new spatial information within short-term memory and in memory consolidation of spatial information [67,68]. In the YM test, in which spontaneous alternation is used to assess spatial working memory [69,70], the abnormalities corresponded to the histopathological abnormalities seen in the CA3 region of the hippocampus.

Our findings are similar to those of previous studies that have shown that sensory and multisensory stimuli can effectively ameliorate the symptoms of AD to improve cognition and behavior. Previous studies have also found sensory stimuli to induce brain neural oscillations and improve brain plasticity, as well as to regulate regional cerebral blood flow [26]. These are important considerations when treating dementia patients, who often have sensory deficits, because of the interactions between sensory and cognitive function. Although clinical studies have been conducted using various sensory stimulation methods, systematic and verified sensory stimulation methods have previously been limited by methodological challenges [71]. Recently, after experimental verification in an animal model of AD, noninvasive gamma frequency sensory stimulation in individuals with mild probable AD was found to improve performance on a delayed recall test, as well as measures of daily activity rhythmicity, compared with a control group in a pilot study [72]. Furthermore, a case series report showed that the long-term use of multisensory gamma stimulation over 1 year contributed to the maintenance of cognitive function and increased mood in patients with MCI or AD [73]. Daily noninvasive gamma frequency sensory stimulation using light or sound was well tolerated, and compliance was equally high in both the control and active group [72,74]. Thus, sensory and multisensory stimulation using the gamma frequency may be a promising approach for treating AD.

This study had several limitations. First, the pathological hallmarks of late-onset AD are Aβ plaques, inflammation, and tau tangles, which occur at different times over the long course of the disease. This study used an animal model of early-stage AD, and the neuropathological analyses focused on Aβ and the cholinergic system. Further studies examining tau protein, microglial activation-related neuroinflammation, and the molecular role of oxidative stress by reactive oxygen species are needed. Moreover, Aβ expression was analyzed using immunohistochemistry in this study. However, it would be helpful if future studies were to confirm the Aβ expression level via an ELISA method that can quantify subtle differences in expression between samples according to anatomical locations in the brain. In the experimental design, we did not include an additional group with normal mice, and we did not perform the stimulation protocol with visual stimulation only. Future studies could include experimental conditions for each type of auditory/visual and combined audiovisual stimulation, including a normal mouse group, to confirm the relationship between single- or multisensory stimulation and cognition. In addition, as noted above, the mouse model in this study was of early-stage sporadic AD. Exploring the therapeutic effects of auditory or audiovisual stimuli in AD requires studies using mice at disease stages corresponding to advanced or mild cognitive impairment.

## 4. Materials and Methods

### 4.1. Experimental Animals and Timeline

Apoe^tm1.1(APOE*4)Adiuj^ (ApoE4-KI, IMSR_JAX: 027894) mice, an experimental model of sporadic AD, were bred, and male, 12 month old mice were used in the experiments. The animals were housed with food and water ad libitum under a 12 h light/dark cycle (8:00 a.m. to 8:00 p.m.) at 23 ± 2 °C and 55% ± 10% humidity. This study followed the guidelines and received the approval of the Institutional Animal Care and Use Committee of Hallym University (Hallym 2018-59, Chunchen, Republic of Korea). The 25 male ApoE4-KI mice were divided into three groups: N, no stimulation (*n* = 5); A, auditory stimulation (*n* = 10); AV, audiovisual stimulation (*n* = 10). Before stimulation, the mice underwent Morris water maze (MYM) training for 5 days. Auditory and audiovisual stimulation were given twice a day for 14 days. Behavioral tests including MYM and Y-maze (YM) tests, were performed before (0 day) and after (7 days, 14 days) auditory and audiovisual stimulation. The mice were euthanized 14 days after sensory stimulation and brain isolation. Mouse brain tissues were then assessed in histopathological and biochemical analyses (Figure 6).

### 4.2. Sensory Stimulation Protocol

Auditory stimulation or combined auditory and visual stimulation was administered to the mice twice a day for 14 days (2 h/day, 14 days) using a soundproof chamber system. Synapse software was used to store and synchronize the digitized data, and RZ6 multiprocessor hardware (Tucker-Davis Technologies, Inc., Alachua, FL, USA) was used to deliver the different stimuli to the mice. For auditory stimulation, the settings were as follows: frequency, 40 Hz; on time, 12.5 ms; time limit, 60 min. The stimulation conditions in the auditory stimulation protocol were set according to the pre-test, and they were based on the experimental methods of previous studies [32,33]. Visual stimulation was delivered using a built-in LUX light driver with an RZ6 multiprocessor, with the following settings: white light LED; wavelength, 400–700 nm; R, 680 Ω; frequency, 40 Hz; on time, 12.5 ms; time limit, 60 min. The auditory and visual stimuli were temporally matched, allowing the simultaneous delivery of auditory and visual stimulation to the AV mice. The stimulation conditions in the combined auditory and visual stimulation protocol were set according to the pre-test, and they were based on the experimental methods of previous studies [33,51].

### 4.3. Behavioral Tests

#### 4.3.1. Morris Water Maze Test

The MWM test was used to assess spatial memory. For water maze training, a circular pool (arena diameter, 100 cm) was filled with water (temperature, 26 °C) made opaque with nontoxic white paint (P0000GSA, Kids mom art, Yangcheon-gu, Seoul, Republic of Korea), and visual cues were placed at the four walls: north, south, east, and west. An escape platform (escape zone diameter, 8 cm) was positioned in the middle of one of the quadrants in the open swimming arena and submerged 1 cm below the water surface. All animals were trained for 120 s twice a day, with at least 4 h between trials. If an animal failed to find the platform within the allotted time, it was usually picked up and placed on the platform for 10 s and then gently dried and warmed. After 5 days of training, the mice performed the MWM at the beginning of the stimulation and 7 and 14 days thereafter. EthoVision XT (Noldus information Technology, Leesburg, VA, USA) video tracking software was used to monitor the mice.

#### 4.3.2. The Y Maze Test

Short-term and spatial working memory was assessed using the YM test, which measures spontaneous alternation. The Y-shaped maze (40 cm length, 12 cm height, 10 cm width) consisted of three arms made of white acrylic resin interconnected at a 120° angle. The mice were placed at the end of the starting arm and allowed to freely explore the three arms of the maze for 5 min. Their behavior was recorded using a video camera (Sony Corp., Tokyo, Minato, Japan) and tracked as described above. The major outcome was the percent spontaneous alternation (alternation index = alternations/maximum alternations × 100).

### 4.4. Histological Analysis

#### 4.4.1. Immunohistochemistry

The left hemisphere of the mice was removed, fixed in 4% paraformaldehyde (BPP-9004, T & I, Chuncheon-si, Gangwon-do, Republic of Korea), and then immersed in a 30% sucrose solution (BS006-5, DUKSAN COMPANY, Ansan-si, Gyeonggi-do, Republic of Korea). The brain tissues were mounted using an embedding compound, frozen at −80 °C, and then sliced (30 μm thickness) using a cryostat microtome (Leica, Wetzlar, Land Hessen, Germany). The frozen sectioned tissue slides were incubated first in 0.3% hydrogen peroxide (H2O2, H0290, Junsei Honsha Co., Ltd., Nihonbashi-honcho, Chuo-ku, Tokyo) for 15 min to block endogenous peroxidase activity and then in 2% horse serum (H0146, Sigma-Aldrich-Merck, Saint Louis, MO, USA) for 60 min. They were then incubated in anti-β-amyloid 1–42 (1:250, ab10148, Abcam, Cambridge, MA, UK) antibody and negative control serum at 4 °C overnight, washed three times for 5 min in 1× phosphate-buffered saline (1× PBS, CBP007B, LPS SOLUTION, Daedeok-gu, Daejeon, Republic of Korea), incubated in horseradish peroxidase (HRP)-conjugated goat anti-rabbit IgG (1:200, 31,402, Invitrogen Corporation, Carlsbad, CA, USA) as the secondary antibody for 120 min at room temperature, and then washed. The tissues were assayed for HRP activity, detected with 3,3′-diaminobenzidine (DAB, 54-10-00, KPL, Gaithersburg, MD, USA), counterstained with Mayer’s hematoxylin, and visualized by light microscopy (Carl Zeiss Microscopy GmbH, Jena, Germany). Amyloid 1–42 positive cells were detected and analyzed using the Image J program (Image J, 1.49v, National Institutes of Health, Bethesda, MD, USA).

#### 4.4.2. Terminal-Deoxynucleotidyl Transferase Mediated Nick end Labeling (TUNEL) Assay

Apoptotic cells in the hippocampus and cortex were assessed using the DeadEnd fluorometric TUNEL system (G3250,Promega, Madison, WI, USA) in tissue slices permeabilized in 0.2% Triton X-100 (TX1061, Georgiachem, Nocross, GA, USA) for 5 min at room temperature. TUNEL labeling was performed by incubating the brain slices with a mixture of recombinant terminal deoxynucleotidyl transferase (rTdT) enzyme and equilibration buffer for 1 h at 37 °C. Negative control rTdT enzyme was replaced with autoclaved, deionized water. The reaction was stopped by placing the slides in rTdT stop buffer for 15 min. The reacted tissue slices were then counterstained with 1 μg of 4′6-diamidino-2-pheny lindole (Molecular Probes, Eugene, OR)/L and rinsed three times in PBS for 5 min. Stained sections were captured using a fluorescence scanning microscope (Olympus, Shinjuku, Tokyo, Japan). The number of TUNEL-positive nuclei was determined using the Image J program (Image J, 1.49v, National Institutes of Health, Bethesda, MD, USA).

### 4.5. Biochemical Analyses

#### 4.5.1. Acetylcholine Assay

The acetylcholine (Ach) level in supernatants prepared from the right hemisphere was measured using a colorimetric choline/acetylcholine assay kit (ab65345, Abcam, Cambridge, MA, UK) following the manufacturer’s guidelines. Samples and standards were reacted with choline and acetylcholinesterase assay buffer for 30 min in the dark and the absorbance at 570 nm was measured on a GloMax Discover microplate reader (Promega, Madison, WI, USA). The Ach concentration was calculated as the difference between total choline and free choline levels (Ach = total choline − free choline).

#### 4.5.2. Apoptotic Protein Assay

Neuronal apoptosis was analyzed by measuring the levels of proapoptotic and antiapoptotic proteins. The right hemisphere was homogenized in ice-cold RIPA buffer (CBR002, LPS SOLUTION, Daedeok-gu, Daejeon, Republic of Korea) containing 1% protease inhibitor (BPI-9200, T & I, Chuncheon-si, Gangwon-do, Republic of Korea) and centrifuged at 13,000× *g* for 10 min at 4 °C. The protein concentration was quantified using a Pierce BCA protein assay kit (23,227, Thermo Fisher Scientific Inc., Waltham, MA, USA). Protein samples (20 μg) were boiled for 10 min, separated on a 10% and 15% SDS-polyacrylamide gel, and transferred onto a PVDF membrane (1620177, Bio-Rad Laboratories, Inc., Hercules, CA, USA). The membrane were blocked with 1× PBS containing 0.1% Tween-20 and 2% bovine serum albumin overnight at 4 °C, and then reacted with primary antibodies against Bax (2772S, 1:1000, Cell Signaling Technology, Danvers, MA, USA), Bcl-2 (4223S, 1:1000, Cell Signaling Technology, Danvers, MA, USA), caspase-3 (9662S, 1:1000, Cell Signaling Technology, Danvers, MA, USA), and β-actin (1:1000, Cell Signaling Technology, Danvers, MA, USA). The secondary antibodies were goat anti-rabbit IgG (ADI-SAB-300-J, 1:2500, Enzo Biochem Inc., Farmingdale, NY, USA) and goat anti-mouse IgG (ADI-SAB-100-J, 1:2500, Enzo Biochem Inc., Farmingdale, NY, USA). The protein bands were imaged using the Image J program (Image J, 1.49v, National Institutes of Health, Bethesda, MD, USA).

### 4.6. Statistical Analysis

The data are presented as the mean ± standard error of the mean (SEM). To check for normality, we performed the Shapiro–Wilk test for every group prior to an analysis of variance (ANOVA) (*p*-value > 0.05). A two-way ANOVA followed by a Bonferroni multiple comparison post hoc test was used to assess sensory stimulation in the different groups and timepoints. A one-way ANOVA followed by a Bonferroni multiple comparison post hoc test was used in statistical comparisons of the three groups. A *p*-value <0.05 was considered to indicate statistical significance. The figures were generated and the statistical analyses were performed using GraphPad Prism 8 software (GraphPad Software Inc, San Diego, CA, USA).

## 5. Conclusions

This study showed that auditory or audiovisual stimulation at 40 Hz can improve cognitive performance and neuropathology in a mouse model of sporadic AD, by decreasing amyloid levels, inhibiting neuronal apoptosis, and enhancing cholinergic transmission in the hippocampus. To our knowledge, this is the first study to demonstrate the therapeutic effects of auditory or audiovisual stimulation in a mouse model of late-onset sporadic AD, the most clinically common type of AD. Further studies of the beneficial effects of auditory or audiovisual intervention for AD, using mice with later-stage disease and focusing on a larger array of histopathological markers, are needed.

## Figures and Tables

**Figure 1 ijms-24-00938-f001:**
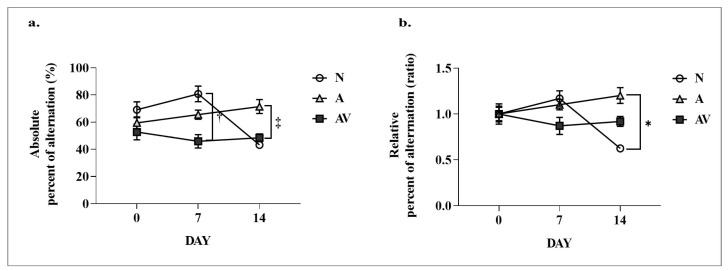
Results of the Y-maze test in ApoE4-KI mice after auditory and audiovisual stimulation. (**a**) absolute percentage alternation; (**b**) relative percentage alternation. ^†^ statistically significant difference between the AV and N groups at 7 days post stimulation (*p* < 0.05); ^‡^ statistically significant difference between the AV and V groups at 14 days post stimulation (*p* < 0.05); * statistically significant difference between A and N groups at 14 days post stimulation (*p* < 0.05). N, no stimulation (*n* = 5); A, auditory stimulation (*n* = 10); AV, audiovisual stimulation (*n* = 10); NS, nonsignificant.

**Figure 2 ijms-24-00938-f002:**
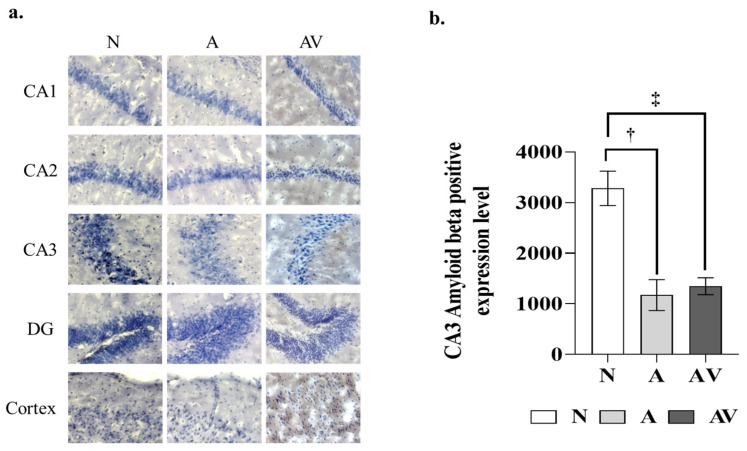
Amyloid-beta expression in ApoE4-KI mice after auditory and audiovisual stimulation. (**a**) Images of Aβ 1–42 positive area; (**b**) Aβ 1–42 positivity in the hippocampal CA3 region. ^†^ statistically significant difference between the A and N groups (*p* < 0.001); ^‡^ statistically significant difference (*p* < 0.001). Aβ, amyloid-beta; CA, cornu ammonis; DG, dentate gyrus; N, no stimulation (*n* = 5); A, auditory stimulation (*n* = 10); AV, audiovisual stimulation (*n* = 10). Eight different tissue slices were used as section replicates, and technical replicates from a single mouse were performed twice.

**Figure 3 ijms-24-00938-f003:**
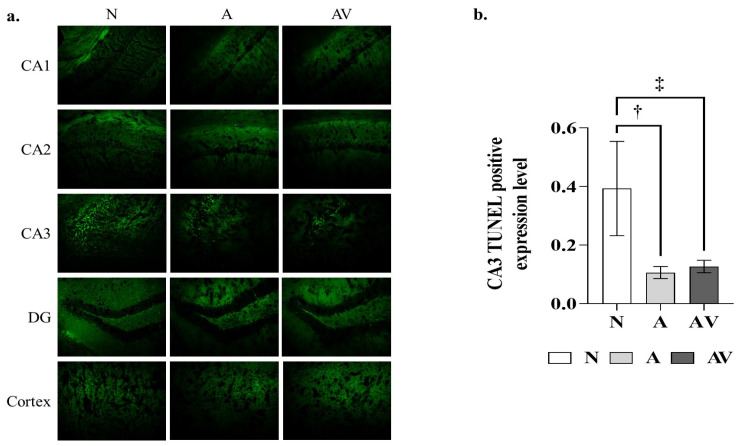
TUNEL positivity in ApoE4-KI mice after auditory and audiovisual stimulation. (**a**) TUNEL positive areas; (**b**) TUNEL positivity in the hippocampal CA3 region. ^†^ statistically significant difference in the hippocampal CA3 region between the A and N groups (*p* < 0.05); ^‡^ statistically significant difference in the hippocampal CA3 region between the AV and N groups (*p* < 0.05). TUNEL, terminal-deoxynucleotidyl transferase mediated nick end labeling; CA, cornu ammonis; DG, dentate gyrus; N, no stimulation (*n* = 5); A, auditory stimulation (*n* = 10); AV, audiovisual stimulation (*n* = 10). Eight different tissue slices were used as section replicates, and technical replicates from a single mouse were performed twice.

**Figure 4 ijms-24-00938-f004:**
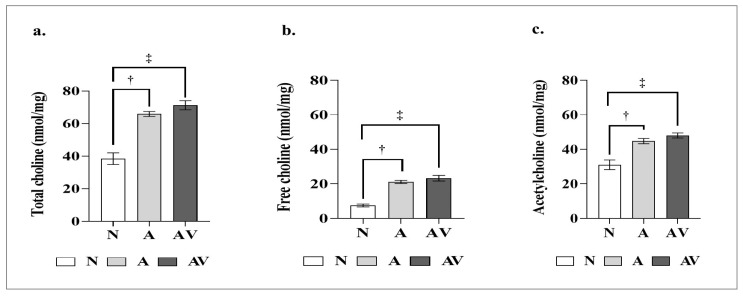
Acetylcholine levels in ApoE4-KI mice after auditory and audiovisual stimulation. (**a**) Total choline; (**b**) free choline; (**c**) acetylcholine. ^†^ statistically significant difference between the A and N groups (*p* < 0.001); ^‡^ statistically significant difference between the AV and N groups (*p* < 0.001). N, no stimulation (*n* = 5); A, auditory stimulation (*n* = 10); AV, audiovisual stimulation (*n* = 10).

**Figure 5 ijms-24-00938-f005:**
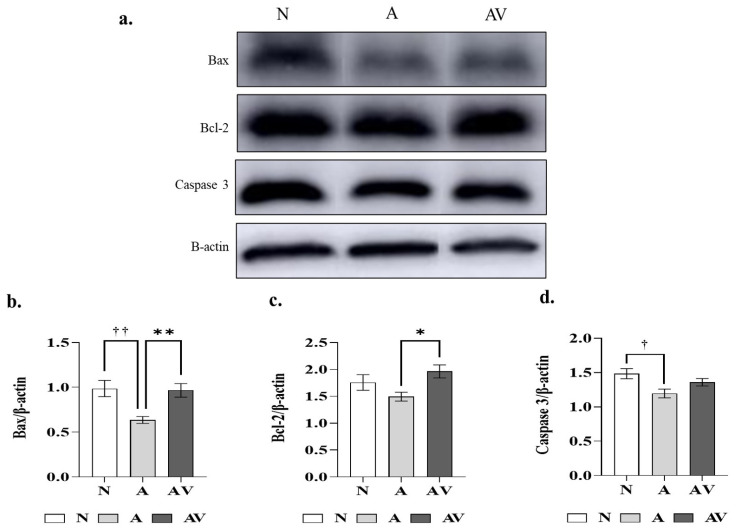
Apoptosis-related proteins in ApoE4-KI mice after auditory and audiovisual stimulation. (**a**) Bax, Bcl-2, caspase-3, and β-actin (control) protein bands; (**b**) relative expression of Bax protein; (**c**) relative expression of Bcl-2 protein; (**d**) relative expression of caspase-3 protein. ^†^ Statistically significant difference between the A and N groups (*p* < 0.05); ^††^ statistically significant difference between the A and N groups (*p* < 0.01); * statistically significant difference between the AV and A groups (*p* < 0.05); ** statistically significant difference between the AV and A groups (*p* < 0.01); ^†^ statistically significant difference between the AV and N groups (*p* < 0.05). N, no stimulation (*n* = 5); A, auditory stimulation (*n* = 10); AV, audiovisual stimulation (*n* = 10). Bax, Bcl-2, and caspase-3 technical replicates were performed twice for each mouse.

**Figure 6 ijms-24-00938-f006:**
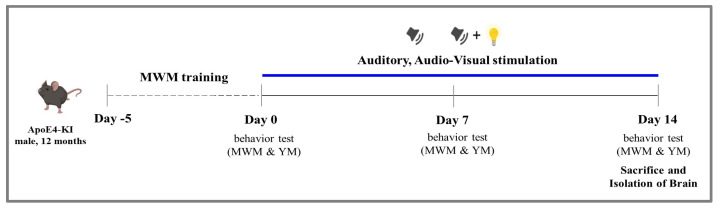
Experimental timelines. After 5 days of training in the MWM, ApoE4-KI mice received auditory and audiovisual stimulation for 14 days (no stimulation, *n* = 5; auditory stimulation, *n* = 10; audiovisual stimulation, *n* = 10). Behavior tests consisting of the MWM and YM were performed before (day 0) and after (day 7, day 14) auditory and audiovisual stimulation. The mice were euthanized 14 days after sensory stimulation. Their brain tissues were then analyzed for histopathological changes. ApoE4-KI, Apoe^tm1.1(APOE*4)Adiuj^; MYM, Morris water maze; YM, Y maze.

**Table 1 ijms-24-00938-t001:** Results of the Y maze test in ApoE4-KI mice after auditory and audiovisual stimulation.

	N	A	AV	*p*-Value		
				Between Times (F Value)	Between Groups (F Value)	Interactions (F Value)
Absolute alternation (%) in YM						
Day 0	69.09 ± 5.09	59.45 ± 6.93	52.74 ± 9.21	<0.001 * (11.59)	ns (2.111)	<0.001 * (4.116)
Day 7	80.77 ± 5.00 ^†^	65.51 ± 5.34	45.9 ± 7.78 ^†^			
Day 14	43.17 ± 1.67	71.46 ± 8.10 ^‡^	48.45 ± 4.53 ^‡^			
Relative alternation (ratio) in YM						
Day 0	1.00 ± 0.07	1.00 ± 0.12	1.00 ± 0.17	<0.05 * (4.055)	ns (1.228)	<0.05 * (3.071)
Day 7	1.17 ± 0.07	1.10 ± 0.09	0.87 ± 0.15			
Day 14	0.62 ± 0.02 ^#^	1.20 ± 0.14 ^#^	0.92 ± 0.09			

Values are expressed as the mean ± SEM. The statistical analyses consisted of a two-way ANOVA followed by Bonferroni’s multiple comparisons test. N, no stimulation (*n* = 5); A, auditory stimulation (*n* = 10); AV, audiovisual stimulation (*n* = 10); MWM, Morris water maze; YM, Y maze; NS, nonsignificant. * Statistically significant in a two-way ANOVA; ^†^ statistically significant difference between the AV and N groups on post-stimulation day 7 (*p* < 0.05); ^‡^ statistically significant difference between the AV and A groups on post-stimulation day 14 (*p* < 0.05); ^#^ statistically significant difference between the A and N groups on post-stimulation day 14 (*p* < 0.05).

**Table 2 ijms-24-00938-t002:** Amyloid-beta expression levels in ApoE4-KI mice after auditory and audiovisual stimulation.

	N	A	AV	*p*-Value
CA1	51.60 ± 23.96	314.10 ± 155.14	384.70 ± 110.97	ns
CA2	113.60 ± 86.41	269.10 ± 99.82	251.30 ± 76.63	ns
CA3	3282.00 ± 338.95	1174.67 ± 457.19 ^†^	1348.00 ± 263.17 ^‡^	<0.001 *
DG	93.00 ± 56.88	217.20 ± 90.11	128.40 ± 39.66	ns
Cortex	322.20 ± 145.70	651.10 ± 227.3	441.40 ± 139.60	ns

Values are expressed as the mean ± SEM. The statistical analysis consisted of a one-way ANOVA followed by Bonferroni’s multiple comparisons test. N, no stimulation (*n* = 5); A, auditory stimulation (*n* = 10); AV, audiovisual stimulation (*n* = 10); NS, nonsignificant. * statistically significant in a one-way ANOVA; ^†^ statistically significant difference in the hippocampal CA3 region between the A and N groups (*p* < 0.001); ^‡^ statistically significant difference in the hippocampal CA3 region between the AV and N groups (*p* < 0.001).

**Table 3 ijms-24-00938-t003:** TUNEL positivity in ApoE4-KI mice after auditory and audiovisual stimulation.

	N	A	AV	*p*-Value
CA1	0.10 ± 0.04	0.05 ± 0.01	0.07 ± 0.03	ns
CA2	0.07 ± 0.03	0.05 ± 0.02	0.12 ± 0.11	ns
CA3	0.39 ± 0.16	0.11 ± 0.03 ^†^	0.13 ± 0.03 ^‡^	<0.01 *
DG	0.11 ± 0.06	0.04 ± 0.01	0.04 ± 0.01	ns
Cortex	0.11 ± 0.06	0.05 ± 0.02	0.07 ± 0.02	ns

Values are expressed as the mean ± SEM. The statistical analyses consisted of a one-way ANOVA followed by Bonferroni’s multiple comparisons test. N, no stimulation (*n* = 5); A, auditory stimulation (*n* = 10); AV, audiovisual stimulation (*n* = 10); NS, nonsignificant. * statistically significant in a one-way ANOVA; ^†^ statistically significant difference in the hippocampal CA3 region between the A and N groups (*p* < 0.05); ^‡^ statistically significant difference in the hippocampal CA3 region between the AV and N groups (*p* < 0.05).

**Table 4 ijms-24-00938-t004:** Acetylcholine levels in ApoE4-KI mice after auditory and audiovisual stimulation.

	N	A	AV	*p*-Value
Total choline (nmol/mg)	149.31 ± 18.79	422.72 ± 28.58 ^†^	465.66 ± 51.82 ^‡^	<0.001 *
Free choline (nmol/mg)	7.47 ± 0.94	21.14 ± 1.43 ^†^	23.28 ± 2.59 ^‡^	<0.001 *
Acetylcholine (nmol/mg)	31.01 ± 3.14	44.77 ± 2.46 ^†^	48.02 ± 2.30 ^‡^	<0.001 *

Values are expressed as the mean ± SEM. The statistical analyses consisted of a one-way ANOVA followed by Bonferroni’s multiple comparisons test. N, no stimulation (*n* = 5); A, auditory stimulation (*n* = 10); AV, audiovisual stimulation (*n* = 10). * statistically significant in a one-way ANOVA; ^†^ statistically significant difference in the hippocampal CA3 region between the A and N groups (*p* < 0.001); ^‡^ statistically significant difference in the hippocampal CA3 region between the AV and N groups (*p* < 0.001).

**Table 5 ijms-24-00938-t005:** Apoptosis-related proteins in ApoE4-KI mice after auditory and audiovisual stimulation.

Apoptosis-Related Proteins	N	A	AV	*p*-Value
Bax/β-actin	0.99 ± 0.10 ^††^	0.64 ± 0.06 ^††,##^	0.97 ± 0.11 ^##^	<0.01 *
Bcl-2/β-actin	1.76 ± 0.16	1.49 ± 0.13 ^#^	1.96 ± 0.19 ^#^	<0.05 *
Caspase-3/β-actin	1.48 ± 0.08 ^†^	1.20 ± 0.10 ^†^	1.36 ± 0.09	<0.05 *

Values are expressed as the mean ± SEM. The statistical analyses consisted of a one-way ANOVA followed by Bonferroni’s multiple comparisons test. N, no stimulation (*n* = 5); A, auditory stimulation (*n* = 10); AV, audiovisual stimulation (*n* = 10). * statistically significant in a one-way ANOVA; ^†^ statistically significant difference between the A and N groups (*p* < 0.05); ^††^ statistically significant difference between the A and N groups (*p* < 0.01); ^#^ statistically significant difference between the AV and A groups (*p* < 0.05); ^##^ statistically significant difference between the AV and A groups (*p* < 0.01).

## Data Availability

The original contributions presented in the study are included in the article; further inquiries can be directed to the corresponding author.

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
