# Peer review of "Auditory or Audiovisual Stimulation Ameliorates Cognitive Impairment and Neuropathology in ApoE4 Knock-In Mice"

_ijms, 2023, doi:10.3390/ijms24020938_

Round 1

Reviewer 1 Report

The manuscript entitled “Auditory or audiovisual stimulation ameliorates cognitive impairment and neuropathology in ApoE4 knock-in mice” addresses the beneficial effect of auditory (A) or audiovisual stimulation (AV) on the cognitive impairment in ApoE4 knock-in mice, a model of early-stage sporadic Alzheimer. Initially, the authors proved that auditory and audiovisual stimulation improved cognitive decline in AD mice. Then, the authors proceeded to some implicated mechanisms. To this end, the authors demonstrated that the enhancement of acetylcholine and suppression of apoptosis were implicated in the observed favorable actions. The current findings are interesting.

Comments:     

1) In the current experimental design, why did not the authors incorporate an additional group (normal mice)? This group would be beneficial for comparing whether the improvement in behavioral outcomes and biochemical determinations approached the normal levels in normal mice without Alzheimer-associated cognitive decline.

2) In the current experimental design, why did not the authors examine the effects of visual stimulation (alone)? This group may be beneficial in delineating the effects/molecular mechanisms of visual stimulation alone. Why did the authors prefer to use combined audiovisual stimulation (AV)?

3) In section 2.2. (Sensory stimulation protocol), how did the authors decide on the auditory stimulation dose in mice? How is the dose/frequency relevant to the human dose? Authors are advised to address this point and add the answers and proper citations to section 2.2.

4) Likewise, in section 2.2. (Sensory stimulation protocol), how did the authors decide on the audiovisual stimulation dose in mice? How is the dose/frequency relevant to the human dose? Authors are advised to address this point and add the answers and proper citations to section 2.2.

5) The authors are advised to add the cat. no. for the used chemicals and antibodies.

6) In immunohistochemistry, did the authors also perform a negative control to ensure the specific binding of the antibody to target protein? Please, add the answer to the comment in section 2.4.1.

7) In the TUNEL assay, did the authors also perform a negative control to ensure the specific binding of the antibody to target protein? Please, add the answer to the comment in section 2.4.2.

8) In the statistical analysis section, did the authors check data normality and homogeneity before proceeding to ANOVA? Authors are advised to address this point and add the answers to section 2.6.

9) In the figure legends, the authors are advised to add the number of animals/replicates from which data were extracted. Authors are advised to address this point and add the answers to all the relevant figure/table legends.

10) In figure 6, the authors are advised to add the number of replicates used in Western blotting. Moreover, were the data extracted from independent samples?

11) To make all figure legends stand-alone, authors are advised to add the full name of the used abbreviations at the end of each legend.

12) Why did not the authors investigate other hallmarks of AD such as tau deposition, neuroinflammation, or oxidative stress markers?

13) In the abstract section, the conclusion should be modified to confirm that the present findings are derived from a mouse mimicking the early-stage sporadic AD.

14) More recent 2022 references are advised to be added to the current manuscript. 

Author Response

Dec 30, 2022 Reviewer 1 Int J Mol Sci Dear Reviewer 1, Please find attached a revised version of our manuscript, “Auditory or audiovisual stimulation ameliorates cognitive impairment and neuropathology in ApoE4 knock-in mice” (ijms-2112594). We thank you for your thoughtful suggestions regarding the original version of our paper; most of the suggested changes have been incorporated into the revision. All revisions are described in detail in the order mentioned in the review, following your comments (in italics). We believe that the revisions have greatly improved the manuscript and hereby submit the revised version for your consideration for publication. Comments to author: The manuscript entitled “Auditory or audiovisual stimulation ameliorates cognitive impairment and neuropathology in ApoE4 knock-in mice” addresses the beneficial effect of auditory (A) or audiovisual stimulation (AV) on the cognitive impairment in ApoE4 knock-in mice, a model of early-stage sporadic Alzheimer. Initially, the authors proved that auditory and audiovisual stimulation improved cognitive decline in AD mice. Then, the authors proceeded to some implicated mechanisms. To this end, the authors demonstrated that the enhancement of acetylcholine and suppression of apoptosis were implicated in the observed favorable actions. The current findings are interesting. We thank the reviewer for these comments and suggestions, which have improved our manuscript. Comments: 1) In the current experimental design, why did not the authors incorporate an additional group (normal mice)? This group would be beneficial for comparing whether the improvement in behavioral outcomes and biochemical determinations approached the normal levels in normal mice without Alzheimer-associated cognitive decline. Thank you for your comment. We agree with your suggestion. We conducted a pre-test using B6 mice as controls to confirm the sensory stimulation settings. However, we found no differences in the behavior tests or neuropathology analyses between mice that received versus did not receive sensory stimulation. Given the small number of mice in each group (2–3 mice), we did not include these data in our analysis. In the future, we plan to include normal mice as controls in an experiment on the various stages of a mouse model of Alzheimer’s disease. We revised the limitations section in the Discussion, as follows. In the experimental design, we did not include an additional group with normal mice, and we did not perform the stimulation protocol with visual stimulation only. Future studies could include experimental conditions for each type of auditory/visual and combined audiovisual stimulation, including a normal mouse group, to confirm the relationship between single-or multi-sensory stimulation and cognition. (page 14, lines 528 – page 14, lines 532) 2) In the current experimental design, why did not the authors examine the effects of visual stimulation (alone)? This group may be beneficial in delineating the effects/molecular mechanisms of visual stimulation alone. Why did the authors prefer to use combined audiovisual stimulation (AV)? Thank you for your comment. Although previous studies have established that hearing loss is an important risk factor in the development of dementia, it is not clear whether intervening to stop hearing loss will also treat progressive cognitive decline. Of various options regarding sensory stimulation, we focused on the relationship between auditory stimulation and dementia, and hypothesized that auditory stimulation could reduce the progression of cognitive decline and ameliorate the symptoms of AD. In addition, we designed our study to assess whether a combination of auditory and visual stimulation could further enhance cognition via multi-sensory integration. However, we did not include a group in which only visual stimulation was applied, and this is a limitation. Thus, we added this information to the limitations section in the Discussion, as follows. In the experimental design, we did not include an additional group with normal mice, and we did not perform the stimulation protocol with visual stimulation only. Future studies could include experimental conditions for each type of auditory/visual and combined audiovisual stimulation, including a normal mouse group, to confirm the relationship between single-or multi-sensory stimulation and cognition. (page 14, lines 528 – page 14, lines 532) 3) In section 2.2. (Sensory stimulation protocol), how did the authors decide on the auditory stimulation dose in mice? How is the dose/frequency relevant to the human dose? Authors are advised to address this point and add the answers and proper citations to section 2.2. We determined our auditory stimulation protocol by conducting a pre-test based on the experimental methods used in previous studies. The stimulation settings used for the animals in our study, i.e., 40 Hz, 2 h per day for 2 weeks, were similar to auditory or audiovisual stimulation settings used in humans. In a clinical pilot study in AD patients, auditory stimulation delivered at 40 Hz (30 min, twice per week for 6 weeks) improved cognitive performance, awareness regarding one’s surroundings, interaction and discussion/storytelling abilities, and alertness. As per the reviewer’s comment, we added and revised the text in the Methods and Discussion, as follows. The stimulation conditions in the auditory stimulation protocol were set according to the pre-test, which were based on the experimental methods of previous studies [32,33]. (page 3, lines 99 – page 3, lines 101) The stimulation settings used for animals in our study, i.e., 40 Hz, 2 h per day for 2 weeks, were similar to the auditory or audiovisual stimulation settings used in previous human studies [41]. In a clinical study, auditory stimulation delivered at 40 Hz (30 min twice a week 6 weeks period) improved cognitive performance in AD patients [31]. (page 12, lines 444 – page 12, lines 448) We have also added the following citations. 31. Clements-Cortes, A.; Ahonen, H.; Evans, M.; Freedman, M.; Bartel, L., Short-Term Effects of Rhythmic Sensory Stimulation in Alzheimer's Disease: An Exploratory Pilot Study. J Alzheimers Dis 2016, 52, (2), 651-60. 32. Juho Lee, S. R., Hyun-Ju Kim, Jieun Jung, Boreom Lee, Tae Kim, 40 Hz acoustic stimulation decreases amyloid beta and modulates brain rhythms in a mouse model of Alzheimer’s disease. bioRxiv 2018. 33. Martorell, A. J.; Paulson, A. L.; Suk, H. J.; Abdurrob, F.; Drummond, G. T.; Guan, W.; Young, J. Z.; Kim, D. N.; Kritskiy, O.; Barker, S. J.; Mangena, V.; Prince, S. M.; Brown, E. N.; Chung, K.; Boyden, E. S.; Singer, A. C.; Tsai, L. H., Multi-sensory Gamma Stimulation Ameliorates Alzheimer's-Associated Pathology and Improves Cognition. Cell 2019, 177, (2), 256-271.e22. 41. Monteiro, F.; Sotiropoulos, I.; Carvalho, Ó.; Sousa, N.; Silva, F. S., Multi-mechanical waves against Alzheimer's disease pathology: a systematic review. Transl Neurodegener 2021, 10, (1), 36. 4) Likewise, in section 2.2. (Sensory stimulation protocol), how did the authors decide on the audiovisual stimulation dose in mice? How is the dose/frequency relevant to the human dose? Authors are advised to address this point and add the answers and proper citations to section 2.2. We established the combined auditory and visual stimulation protocol via a pre-test based on the experimental methods used in previous studies. The stimulation settings used for the animals in our study, i.e., 40 Hz, 2 h per day for 2 weeks, were similar to the auditory or audiovisual stimulation settings used in human studies. Preliminary evidence suggested that audiovisual gamma flicker stimulation (40 Hz, 1 h daily for 4 or 8 weeks) strengthened functional connectivity and altered brain immune factors in patients with mild cognitive impairment. As per the reviewer’s comment, we added and revised the text in the Methods and Discussion, as follows. The stimulation conditions in the combined auditory and visual stimulation protocol were set according to the pre-test, which were based on the experimental methods of previous studies [33,40]. (page 3, lines 105 – page 3, lines 108) The stimulation settings used for animals in our study, i.e., 40 Hz, 2 h per day for 2 weeks, were similar to the auditory or audiovisual stimulation settings used in previous human studies [41]. In a clinical study, auditory stimulation delivered at 40 Hz (30 min twice a week 6 weeks period) improved cognitive performance in AD patients [31]. In a preliminary clinical trial, gamma flicker using audiovisual stimulation (40 Hz, 1 h daily 4 or 8 weeks) strengthened functional connectivity and altered brain immune factors in patients with mild cognitive impairment related to AD, as evidenced by resting state EEG, neural circuits via default mode connectivity in fMRI, immune signaling via changes in cytokines, and immune factors in the CSF [53]. (page 12, lines 444 – page 13, lines 455) We have also added the following citations. 33. Martorell, A. J.; Paulson, A. L.; Suk, H. J.; Abdurrob, F.; Drummond, G. T.; Guan, W.; Young, J. Z.; Kim, D. N.; Kritskiy, O.; Barker, S. J.; Mangena, V.; Prince, S. M.; Brown, E. N.; Chung, K.; Boyden, E. S.; Singer, A. C.; Tsai, L. H., Multi-sensory Gamma Stimulation Ameliorates Alzheimer's-Associated Pathology and Improves Cognition. Cell 2019, 177, (2), 256-271.e22. 40. Adaikkan, C.; Middleton, S. J.; Marco, A.; Pao, P. C.; Mathys, H.; Kim, D. N.; Gao, F.; Young, J. Z.; Suk, H. J.; Boyden, E. S.; McHugh, T. J.; Tsai, L. H., Gamma Entrainment Binds Higher-Order Brain Regions and Offers Neuroprotection. Neuron 2019, 102, (5), 929-943.e8. 5) The authors are advised to add the cat. no. for the used chemicals and antibodies. We have added the catalog number in the Methods. 6) In immunohistochemistry, did the authors also perform a negative control to ensure the specific binding of the antibody to target protein? Please, add the answer to the comment in section 2.4.1. We have revised the text in the Methods, as follows. They were then incubated in anti-β-amyloid 1-42 (1:250, ab10148, Abcam, Cambridge, MA, UK) antibody and negative control serum at 4°C overnight, washed three times for 5 min in 1× phosphate-buffered saline (1× PBS, CBP007B, LPS SOLUTION, Daedeok-gu, Daejeon, Republic of Korea), incubated in horseradish peroxidase (HRP)-conjugated goat anti-rabbit IgG (1:200, 31402, Invitrogen Corporation, Carlsbad, CA, USA) as the secondary antibody for 120 min at room temperature, and then washed. (page 4, lines 153 – page 4, lines 159) 7) In the TUNEL assay, did the authors also perform a negative control to ensure the specific binding of the antibody to target protein? Please, add the answer to the comment in section 2.4.2. We have added text in the Methods, as follows. Negative control rTdT enzyme was replaced with autoclaved, deionized water. (page 4, lines 171 – page 4, lines 172) 8) In the statistical analysis section, did the authors check data normality and homogeneity before proceeding to ANOVA? Authors are advised to address this point and add the answers to section 2.6. To check for normality, we performed the Shapiro-Wilk test for every group prior to conducting an ANOVA. The p-value was greater than 0.05, and the data were normally distributed. We have added new text in the Results, as follows. To check for normality, we performed the Shapiro-Wilk test for every group prior to an analysis of variance (ANOVA) (p-value > 0.05). (page 5, lines 210 – page 5, lines 212) 9) In the figure legends, the authors are advised to add the number of animals/replicates from which data were extracted. Authors are advised to address this point and add the answers to all the relevant figure/table legends. As per the reviewer’s comment, we included the number of animals in all of the relevant figure/table legends. For the biological replicates, the number of animals used in the experimental groups was as follows: No stimulation group (n = 5), Auditory group (n = 10), Audiovisual group (n = 10). For the immunohistochemistry and TUNEL assay, eight different tissue slices were used as section-replicates, and we performed technical replicates twice for each mouse. Accordingly, we have revised the legends of the Tables and Figures. 10) In figure 6, the authors are advised to add the number of replicates used in Western blotting. Moreover, were the data extracted from independent samples? The technical replicates for Bax, Bcl-2, and Caspase 3 were performed twice. The samples used for western blotting were analyzed independently for each group. For the biological replicates, the number of animals used in the experimental groups was as follows: No stimulation group (n = 5), Auditory group (n = 10), Auditory visual group (n = 10). Accordingly, we have revised the legend for Figure 6. 11) To make all figure legends stand-alone, authors are advised to add the full name of the used abbreviations at the end of each legend. As per the reviewer’s comment, we have revised the legends for Figures 1–6. 12) Why did not the authors investigate other hallmarks of AD such as tau deposition, neuroinflammation, or oxidative stress markers? The reviewer raises an important point. The pathological hallmarks of late-onset AD are Aβ plaques, inflammation, and tau tangles, and these occur at different times over the long course of the disease. In this study, we used an animal model of early-stage AD, and the neuropathological analysis focused on Aβ and the cholinergic system. Further studies examining tau protein, microglial activation-related neuroinflammation, and the molecular role of oxidative stress by reactive oxygen species are needed. This is a limitation of our study, and we revised the text in the Discussion accordingly, as follows. First, the pathological hallmarks of late-onset AD are Aβ plaques, inflammation, and tau tangles, which occur at different times over the long course of the disease. This study used an animal model of early-stage AD, and the neuropathological analyses focused on Aβ and the cholinergic system. Further studies examining tau protein, microglial activation-related neuroinflammation, and the molecular role of oxidative stress by reactive oxygen species are needed. (page 14, lines 518 – page 14, lines 524) 13) In the abstract section, the conclusion should be modified to confirm that the present findings are derived from a mouse mimicking the early-stage sporadic AD. As per the reviewer’s comment, we revised the text in the Abstract, as follows. In a mouse model of early-stage sporadic AD, auditory or audiovisual stimulation improved cognitive performance and neuropathology. (page 1, lines 23 – page 1, lines 25) 14) More recent 2022 references are advised to be added to the current manuscript. We have added new text to the Discussion, as follows. Although clinical studies have been conducted using various sensory stimulation methods, systematic and verified sensory stimulation methods have previously been limited by methodological challenges [71]. Recently, after experimental verification in an animal model of AD, noninvasive gamma frequency sensory stimulation in individuals with mild probable AD was found to improve performance on a delayed recall test, as well as measures of daily activity rhythmicity, compared with a control group in a pilot study [72]. Furthermore, a case series report showed that the long-term use of multi-sensory gamma stimulation over one year contributed to the maintenance of cognitive function and increased mood in patients with MCI or AD [73]. Daily non-invasive gamma frequency sensory stimulation using light or sound was well tolerated, and compliance was equally high in both the control and active group [72, 74]. Thus, sensory and multisensory stimulation using the gamma frequency may be a promising approach for treating AD. (page 14, lines 506 – page 14, lines 517) We have also added the following citations, including recent references from 2022. 71. Pinto, J.O,; Dores, A.R.; Geraldo, A.; Peixoto, B.; Barbosa, F, Sensory stimulation programs in dementia: a systematic review of methods and effectiveness. Expert Rev Neurother 2020, 21, (12),1229-1247. 72. Chan, D.; Suk, H.J.; Jackson, B.L.; Milman, N.P.;Stark, D.; Klerman, B.E, Gamma frequency sensory stimulation in mild probable Alzheimer's dementia patients: Results of feasibility and pilot studies. PLoS One 2022, 1, 17, (12), e0278412. 73. Amy, C.C.; Lee, B, Long-Term Multi-Sensory Gamma Stimulation of Dementia Patients: A Case Series Report. Int J Environ Res Public Health 2022, 23,19, (23),15553. 74. Khachatryan, E.; Wittevrongel, B.; Reinartz, M.; Dauwe. L.; Carrette, E.; Meurs, A, Cognitive tasks propagate the neural entrainment in response to a visual 40 Hz stimulation in humans. Front Aging Neurosci. 2022, 6,14,1010765. We have addressed all of the issues raised by the reviewers. We are grateful for the constructive comments made during the review process. We believe that our paper has been improved by implementing these suggestions. Yours faithfully, Jong-Hee Sohn, M.D. Ph.D. Department of Neurology, Chuncheon Sacred Heart Hospital, Hallym University College of Medicine, 77 Sakju-ro, Chuncheon-si, Gangwon-do, 24253, Republic of Korea Tel: +82-33-252-9970, Fax: +82-33-241-8063 E-mail: deepfoci@hallym.or.kr

Reviewer 2 Report

In the manuscript by Jung et al., the authors compared the effects of auditory and audiovisual stimulation on the cognitive function and AD-related pathologies in APOE4-KI mice. They found both auditory and audiovisual stimulation could improve cognitive performance and alleviate neuropathology in APOE4-KI mice, whereas audiovisual stimulation didn’t show additional effects compared to auditory stimulation. The experiments are well-designed and the results are convincing. There are still several aspects need to be improved before being considered for publication.

1. In the manuscript, the authors showed MWM results in tables, which is difficult for the readers to catch the information. The authors are suggested to show the important results using bar graphs and move the tables to the supplementary file.

2. The authors performed Ab immunostaining to quantify the changes of amyloid pathology in mouse brain. Due to the lack of extracellular amyloid plaque deposition, only a few intracellular Ab staining shown in the representative images. the authors should show enlarged representative images to make it clearer. And also, immunostaining might not be sensitive enough to detect the Ab changes in other regions. To make the results more convincing, the authors should also measure Ab levels using ELISA to confirm the staining results.

3. From the literature and the findings in this manuscript, it seems sensory stimulus can be promising tools in AD treatment. It will be more informative if the authors can discuss more about the limitations or side effects of sensory stimulus as a new form of AD therapeutic strategy. 

Author Response

Dec 30, 2022 Reviewer 2 Int J Mol Sci Dear Reviewer 2, Please find attached a revised version of our manuscript, “Auditory or audiovisual stimulation ameliorates cognitive impairment and neuropathology in ApoE4 knock-in mice” (ijms-2112594). We thank you for your thoughtful suggestions regarding the original version of our paper; most of the suggested changes have been incorporated into the revision. All revisions are described in detail in the order mentioned in the review, following your comment (in italics). We believe that the revisions have greatly improved the manuscript and hereby submit the revised version for your consideration for publication Comments to author: In the manuscript by Jung et al., the authors compared the effects of auditory and audiovisual stimulation on the cognitive function and AD-related pathologies in APOE4-KI mice. They found both auditory and audiovisual stimulation could improve cognitive performance and alleviate neuropathology in APOE4-KI mice, whereas audiovisual stimulation didn’t show additional effects compared to auditory stimulation. The experiments are well-designed and the results are convincing. There are still several aspects need to be improved before being considered for publication. We thank the reviewer for these comments and suggestions, which have improved our manuscript. 1. In the manuscript, the authors showed MWM results in tables, which is difficult for the readers to catch the information. The authors are suggested to show the important results using bar graphs and move the tables to the supplementary file. In the MWM test, we measured the distance moved, velocity (cm/s), time spent moving (%), time spent not moving (%), time in each quadrant (%), absolute escape latency, and relative escape latency (ratio). Of the various parameters measured in the MWM test, the relative escape latency of the A group was less than that of the N group, but this difference was not significant. There were no statistically significant differences in the other parameters of the MWM test. Initially, we included the results for all of the parameters of the MWM test in Supplementary Table S1 and Figure S1, and some of this information was also given in Table 1. In the revised manuscript, we have removed the redundant data from Table 1. We have also revised Table 1. 2. The authors performed Ab immunostaining to quantify the changes of amyloid pathology in mouse brain. Due to the lack of extracellular amyloid plaque deposition, only a few intracellular Ab staining shown in the representative images. the authors should show enlarged representative images to make it clearer. And also, immunostaining might not be sensitive enough to detect the Ab changes in other regions. To make the results more convincing, the authors should also measure Ab levels using ELISA to confirm the staining results. We agree with your suggestions. We have revised Figure 3 by increasing the magnification of the image to make it clearer. Our study showed that auditory or audiovisual stimulation improved cognitive performance and neuropathology in a mouse model of early-stage sporadic AD. We focused on the positive effects on cognitive performance and neuropathological findings, and confirmed our findings by analyzing Aβ expression via immunohistochemistry according to anatomical locations in the brain. We used the left hemisphere for western blotting samples and the right hemisphere for immunohistochemistry. As stated by the reviewer, ELISA is an experimental technique that can quantify subtle differences in expression between samples. Further studies are needed confirm Aβ expression levels according to anatomical location such as the hippocampus and cortex. In the revised manuscript, we modified Figure 3 and added limitations to the Discussion, as follows. Also, Aβ expression was analyzed using immunohistochemistry in this study. However, it would be helpful if future studies were to confirm the Aβ expression level via an ELISA method that can quantify subtle differences in expression between samples according to anatomical locations in the brain. (page 14, lines 524 – page 14, lines 528) 3. From the literature and the findings in this manuscript, it seems sensory stimulus can be promising tools in AD treatment. It will be more informative if the authors can discuss more about the limitations or side effects of sensory stimulus as a new form of AD therapeutic strategy. Sensory-based interventions, including sensory stimulation, are important for dementia patients because they may address sensory deficits and enhance interactions between sensory and cognitive processes. Although clinical studies using various sensory stimulation methods have been conducted, systematic and verified sensory stimulation methods have been limited by various methodological challenges. Previous studies have shown that sensory and multisensory stimuli can effectively ameliorate the symptoms of AD in terms of cognition and behavior. Such stimuli have also been found to induce neural oscillations, improve brain plasticity, and regulate regional cerebral blood flow. Recently, after experimental verification in an animal model of AD, a pilot study showed that non-invasive gamma frequency sensory stimulation in individuals with mild probable AD improved performance on a delayed recall test, as well as measures of daily activity rhythmicity compared with a control group. Also, a case series report showed that the long-term use of multi-sensory gamma stimulation over one year contributed to maintained cognition and increased mood for patients with mild cognitive impairment or AD. These daily non-invasive gamma frequency sensory stimulation sessions, which used light or sound, were well tolerated and compliance was equally high in the control and active group. Thus, sensory and multisensory gamma frequency stimulation appear to be promising therapeutic strategies for dementia patients. We have revised the text in the Discussion, as follows. Our findings are similar to those of previous studies that have shown that sensory and multisensory stimuli can effectively ameliorate the symptoms of AD to improve cognition and behavior. Previous studies have also found sensory stimuli to induce brain neural oscillations and improve brain plasticity, as well as to regulate regional cerebral blood flow [26]. are important considerations when treating dementia patients, who often have sensory deficits, because of the interactions between sensory and cognitive function. Although clinical studies have been conducted using various sensory stimulation methods, systematic and verified sensory stimulation methods have previously been limited by methodological challenges [71]. Recently, after experimental verification in an animal model of AD, noninvasive gamma frequency sensory stimulation in individuals with mild probable AD was found to improve performance on a delayed recall test, as well as measures of daily activity rhythmicity, compared with a control group in a pilot study [72]. Furthermore, a case series report showed that the long-term use of multi-sensory gamma stimulation over one year contributed to the maintenance of cognitive function and increased mood in patients with MCI or AD [73]. Daily non-invasive gamma frequency sensory stimulation using light or sound was well tolerated, and compliance was equally high in both the control and active group [72, 74]. Thus, sensory and multisensory stimulation using the gamma frequency may be a promising approach for treating AD. (page 13, lines 500 – page 14, lines 517) We have also added the following citations. 71. Pinto, J.O,; Dores, A.R.; Geraldo, A.; Peixoto, B.; Barbosa, F, Sensory stimulation programs in dementia: a systematic review of methods and effectiveness. Expert Rev Neurother 2020, 21, (12),1229-1247. 72. Chan, D.; Suk, H.J.; Jackson, B.L.; Milman, N.P.;Stark, D.; Klerman, B.E, Gamma frequency sensory stimulation in mild probable Alzheimer's dementia patients: Results of feasibility and pilot studies. PLoS One 2022, 1, 17, (12), e0278412. 73. Amy, C.C.; Lee, B, Long-Term Multi-Sensory Gamma Stimulation of Dementia Patients: A Case Series Report. Int J Environ Res Public Health 2022, 23,19, (23),15553. 74. Khachatryan, E.; Wittevrongel, B.; Reinartz, M.; Dauwe. L.; Carrette, E.; Meurs, A, Cognitive tasks propagate the neural entrainment in response to a visual 40 Hz stimulation in humans. Front Aging Neurosci. 2022, 6,14,1010765. We have addressed all of the issues raised by the reviewers. We are grateful for the constructive comments made during the review process. We believe that our paper has been improved by implementing these suggestions. Yours faithfully, Jong-Hee Sohn, M.D. Ph.D. Department of Neurology, Chuncheon Sacred Heart Hospital, Hallym University College of Medicine, 77 Sakju-ro, Chuncheon-si, Gangwon-do, 24253, Republic of Korea Tel: +82-33-252-9970, Fax: +82-33-241-8063 E-mail: deepfoci@hallym.or.kr

Round 2

Reviewer 2 Report

The manuscript has been sufficiently improved to warrant publication in IJMS.